# Genotyping-by-Sequencing Analysis Reveals Associations between Agronomic and Oil Traits in Gamma Ray-Derived Mutant Rapeseed (*Brassica napus* L.)

**DOI:** 10.3390/plants13111576

**Published:** 2024-06-06

**Authors:** Woon Ji Kim, Baul Yang, Dong-Gun Kim, Sang Hoon Kim, Ye-Jin Lee, Juyoung Kim, So Hyeon Baek, Si-Yong Kang, Joon-Woo Ahn, Yu-Jin Choi, Chang-Hyu Bae, Kanivalan Iwar, Seong-Hoon Kim, Jaihyunk Ryu

**Affiliations:** 1Advanced Radiation Technology Institute, Korea Atomic Energy Research Institute, Jeongeup 56212, Republic of Korea; wjkim0101@kaeri.re.kr (W.J.K.); dgkim@kaeri.re.kr (D.-G.K.); shkim80@kaeri.re.kr (S.H.K.); yjinlee@kaeri.re.kr (Y.-J.L.); jykim83@kaeri.re.kr (J.K.); joon@kaeri.re.kr (J.-W.A.); 2Imsil Cheese & Food Research Institute, Imsil-gun 55918, Republic of Korea; byang@icf.re.kr (B.Y.); yj1730@icf.re.kr (Y.-J.C.); 3Department of Plant Production Sciences, Graduate School, Sunchon National University, Suncheon 57922, Republic of Korea; baeksh@scnu.ac.kr (S.H.B.); chbae@scnu.ac.kr (C.-H.B.); 4Department of Horticulture, College of Industrial Sciences, Kongju National University, Yesan 32439, Republic of Korea; sykang@kongju.ac.kr; 5National Agrobiodiversity Center, National Institute of Agricultural Sciences, Rural Development Administration, Jeonju 5487, Republic of Korea; kani05@korea.kr (K.I.); shkim0819@korea.kr (S.-H.K.)

**Keywords:** rapeseed, single nucleotide polymorphisms, genotyping-by-sequencing, genomewide association study, gene ontology

## Abstract

Rapeseed (*Brassica napus* L.) holds significant commercial value as one of the leading oil crops, with its agronomic features and oil quality being crucial determinants. In this investigation, 73,226 single nucleotide polymorphisms (SNPs) across 95 rapeseed mutant lines induced by gamma rays, alongside the original cultivar (‘Tamra’), using genotyping-by-sequencing (GBS) analysis were examined. This study encompassed gene ontology (GO) analysis and a genomewide association study (GWAS), thereby concentrating on agronomic traits (e.g., plant height, ear length, thousand-seed weight, and seed yield) and oil traits (including fatty acid composition and crude fat content). The GO analysis unveiled a multitude of genes with SNP variations associated with cellular processes, intracellular anatomical structures, and organic cyclic compound binding. Through GWAS, we detected 320 significant SNPs linked to both agronomic (104 SNPs) and oil traits (216 SNPs). Notably, two novel candidate genes, Bna.A05p02350D (SFGH) and Bna.C02p22490D (MDN1), are implicated in thousand-seed weight regulation. Additionally, Bna.C03p14350D (EXO70) and Bna.A09p05630D (PI4Kα1) emerged as novel candidate genes associated with erucic acid and crude fat content, respectively. These findings carry implications for identifying superior genotypes for the development of new cultivars. Association studies offer a cost-effective means of screening mutants and selecting elite rapeseed breeding lines, thereby enhancing the commercial viability of this pivotal oil crop.

## 1. Introduction

Rapeseed (*Brassica napus* L.), an interspecific amphidiploid hybrid with a chromosome count of 2n = 38, originates from the natural hybridization between *B. rapa* and *B. oleracea* [1,2]. This versatile crop is esteemed for its capacity to yield substantial quantities of oil, thus making it highly prized as a nutritious vegetable oil globally [2]. Its applications span various industries including food, animal feed, energy, and chemicals [3,4,5]. Consequently, breeders have persistently endeavored to develop new rapeseed cultivars with enhanced agronomic traits such as increased yields, disease resistance, and desirable oil characteristics, including modified fatty acid profiles and crude fat content. However, genetic resources for rapeseed in Korea pertinent to its agronomic and oil-related traits are limited [6]. Given this scarcity of genetic diversity, mutagenesis presents a promising strategy for generating novel genetic variations. This approach facilitates the acquisition of desired traits in rapeseed, such as improved yield and altered fatty acid composition [3,6].

The weight of rapeseed seeds holds immense importance as a crucial factor influencing overall yield, thus playing a pivotal role in plant evolution and crop improvement strategies [6,7]. Small-sized seeds possess a higher potential for dispersal, while larger seeds demonstrate increased adaptability to various biotic and abiotic stressors [2,8]. Moreover, seedlings emerging from larger seeds often exhibit enhanced competitive survival rates compared to those from smaller seeds, thus underscoring the significance of seed weight in determining plant fitness and survival [2,7,8]. Many cultivated crops exhibit larger seed sizes compared to their wild relatives [7,8,9], thus emphasizing the importance of identifying genes associated with thousand-seed weights in agricultural research and breeding efforts.

The economic viability of industrial applications involving rapeseed oil heavily relies on its quality. The nutritional properties of rapeseed oil, which are crucial for its market value, primarily stem from its fatty acid compositions synthesized via biochemical pathways involving acetyl-CoA and NADPH [6,10]. Historically, rapeseed oil, which is renowned for its high erucic acid content, has faced limited utilization due to its adverse effects on animal cardiac health [4,11,12]. However, modern breeding endeavors in edible oil production have shifted towards developing canola-type rapeseed varieties with “double low” seeds, thereby featuring reduced erucic acid and glucosinolate levels [11]. The fatty acid profile of canola-type rapeseed oil, comprising 7% palmitic acid, 2% stearic acid, 61% oleic acid, 11% linoleic acid, and 21% linolenic acid, has been deemed nutritionally optimal. High-oleic acid oil (>70%) derived from canola-type rapeseed has gained popularity as a healthy and stable cooking oil [5,11]. Conversely, rapeseed oil with high-erucic acid content finds applications in various industries such as polyethylene films, biodegradable plastics, biodiesel production, printing, and steel manufacturing [11,13]. In the crushing industry, the primary value is derived from the oil content of oilseed rapeseeds, despite the protein’s significance for animal feed [4,5,11]. This aspect is particularly crucial for the biodiesel and cooking oil sectors, where production cost optimization is paramount [5,13]. Given the expanding utilization of rapeseed oil as a renewable feedstock, increasing seed oil content holds significant economic implications. Hence, optimizing the fatty acid composition of rapeseed oil stands as a central objective in numerous breeding programs.

Mutation breeding involves the deliberate use of physical and/or chemical mutagens to induce genetic alterations in plants, thus ultimately leading to the development of desirable characteristics that are suitable for commercial purposes [6,14]. This process requires the careful selection of mutations that effectively modify both agronomic traits and oil characteristics, which is achievable through various mutagenesis techniques [9,15]. Of the nearly 3000 plant mutant varieties released worldwide, more than 60% were created by physical radiation (γ-rays or X-rays) [16]. Among these techniques, gamma ray irradiation stands out as one of the most widely employed methods in plant mutation breeding. Gamma rays are a form of ionizing radiation that causes double-strand breaks in DNA, thus resulting in base substitutions, indels, copy number variations, and presence/absence variations [17]. Gamma ray irradiation has proven successful in inducing translocations in amphiploid species, thereby introducing desirable genes that contribute to enhanced seed yields, the development of semidwarf varieties, and broader resistance to diseases [6,14,15]. Furthermore, mutagenesis through gamma irradiation has been found to stimulate genetic recombination processes, thereby broadening the spectrum of mutations induced and augmenting the overall efficacy of the technique [6,18].

The advent of next-generation sequencing (NGS) technology has revolutionized the sequencing of plant genomes, thus facilitating the direct detection of single nucleotide polymorphisms (SNPs) [6]. This advancement has greatly contributed to the development of cultivars with desired traits in plant breeding. Genotyping-by-sequencing (GBS) analysis is a technique that simplifies genomic complexity by fragmenting the genome into smaller pieces using restriction enzymes, which are then sequenced on short-read platforms [19,20]. With the increasing availability of complete genome sequences and SNP arrays, association mapping has emerged as a robust approach for elucidating genetic characteristics, thereby enhancing the precision of quantitative trait locus (QTL)-based position estimations [20,21]. Association mapping has proven particularly advantageous in circumventing the limitations of traditional QTL mapping, notably due to the vast number of SNP markers identified by NGS technologies. The recent completion of the *B. napus* genome has facilitated direct comparisons between documented complex traits identified through mapping studies [5,6,9]. GBS-enabled SNP identification not only facilitates the analysis of genetic diversity but also streamlines the integration of genomewide association studies (GWASs) into comprehensive research projects [19,22]. GWASs serve as a potent tool for identifying QTLs and genes in various crops, including rapeseed, soybean, and rice, thereby advancing our understanding of complex trait inheritance and enabling targeted breeding efforts [8,20,22,23].

We have cultivated mutant rapeseed lines through gamma radiation mutation, with each exhibiting diverse agronomic characteristics such as plant height, ear length, thousand-seed weight, and seed yield, alongside variations in oil traits, including fatty acid compositions and crude fat content. The primary objectives of this study were to analyze the SNPs present in 95 rapeseed mutant lines and to pinpoint candidate genes associated with both agronomic and oil traits using GWASs.

## 2. Results

### 2.1. Assessment of Agronomic Traits

Ninety-five rapeseed mutant lines, along with the original cultivar ‘Tamra’, were assessed for their agronomic traits and fatty acid content (Appendix A). The rapeseed mutant population exhibited considerable variability in the traits measured over 2 years. The distribution of the four agronomic traits of plant height, ear length, thousand-seed weight, and seed yield is shown in Table 1 and Figure 1. The plant height of the original cultivar measured 163.5 cm. Among the rapeseed mutant lines, the plant height ranged from 134.0 cm (Tr2-2) to 175.0 cm (Tr25-14), with a mean of 157.7 cm. The ear length of the original cultivar was recorded at 55.5 cm, while the ear length across all mutant lines varied from 35.0 cm (Tr8-3-1) to 77.5 cm (Tr38-7), with a mean of 59.8 cm. Significant differences were observed in both the thousand-seed weight and seed yield among all mutant lines. The original cultivar exhibited a thousand-seed weight of 3.8 g, with the highest recorded in the Tr6-11-1 line (5.4 g) and the lowest in the Tr138-L line (2.8 g). Regarding the seed yield, the original cultivar yielded 309 kg/10a, with the highest seed yield (398 kg/10a) observed in the Tr38-4 line and the lowest (144 kg/10a) in the Tr14-3 line. The coefficients of variation were lowest for plant height at 4.8% and highest for seed yield at 26.2%.

### 2.2. Fatty Acid Compositions and Crude Fat Contents

The fatty acid profiles and crude fat content of the original cultivar ‘Tamra’ and 95 rapeseed mutant lines are presented in Table 2 and Figure 2. The original cultivar exhibited a composition of 5.5% palmitic (C16:0), 1.9% stearic (C18:0), 65.5% oleic (C18:1), 20.5% linoleic (C18:2), 5.6% linolenic (C18:3), and 1.0% eicosenoic acid (C20:1), with no detectable levels of palmitoleic acid (C16:1) or erucic acid (C22:1). However, palmitoleic acid was detected in 50 mutant lines, thus ranging from 0.1% to 0.76%. The stearic acid content in the mutant lines varied from 0.2% to 4.4%, thus averaging 1.7%. Significant variations in the oleic acid content were observed, thus ranging from 32.4% to 81.7% among the mutant lines, with the highest content in the Tr4 mutant line. Additionally, nine mutant lines exhibited oleic acid compositions exceeding 70%. The highest linoleic acid content was recorded in the Tr138-15 line (33.5%), while the lowest was in the Tr1-1 line (7.9%). The linolenic acid content ranged from 4.0% in the Tr65-7 line to 13.7% in the Tr5-3 line. Eicosenoic acid was absent in seven mutant lines, with the remaining lines showing compositions ranging from 0.4% to 13.1%, with the highest in the Tr14-1 line. Erucic acid was detected in 43 mutant lines, ranging from 0.2% to 29.0%, with relatively high levels (>20%) observed in the Tr14-1 and Tr14-5 lines. The original cultivar exhibited a crude fat content of 40.16 mg/100 g, with mutant lines ranging from 21.31 to 40.98 mg/100 g. Nine mutant lines showed crude fat content levels similar to ‘Tamra’, while Tr14-5, Tr18-2, Tr25-1, and Tr25-2 contained lower levels. Compared to a coefficient of variation (CV) of 10.4% for the crude fat content, the fatty acid composition showed considerably greater variation, with the CV ranging from 18.5 to 128.8%.

The correlations between each fatty acid composition are shown in Table 3. C20:1 and C22:1 had the highest positive correlation, with *r* = 0.871 (*p* ≤ 0.01), while C18:1 and C20:1 had the lowest negative correlation, with *r* = −0.639 (*p* ≤ 0.01).

### 2.3. GBS Analysis of Rapeseed Mutant Lines

A GBS library containing 96 rapeseed genotypes, comprising 95 newly developed mutant lines and the original cultivar, underwent sequencing using the Illumina HiSeq X ten platform (Illumina, Madison, WI, USA). The sequencing results are summarized in Table 4. In total, 715 million reads were generated, thus amounting to 108,004,241,578 nucleotides (108 Gb), with an average of 7.45 million reads per genotype. Following the removal of low-quality sequences, 655,243,166 clean reads were retained, thus averaging 6.8 million reads per genotype. The length of the clean reads ranged from 73,633,121 base pairs (bp) to 5,305,962,156 bp, with an average read length of 777,311,365 bp (Appendix A). Across all lines, a total of 651,288,040 reads were successfully mapped, thus averaging 6,784,250 reads per sample. The mapped read rates (%) ranged from 98.98% to 99.46%, with an average of 99.39% of filtered reads mapped to the reference genome sequence. The total length of the mapped region was 3,163,629,539 bp, thus averaging 32,954,474 bp per sample and covering approximately 3.57% of the reference genome sequence.

### 2.4. Gene Ontology (GO) Analysis of Genes with Ploymorphic SNPs

GO enrichment analysis was performed to functionally classify the genes mutated by gamma ray irradiations in the mutant lines. Genes carrying polymorphic SNPs (*p* < 0.05) were subjected to analysis. These genes were categorized into three main functional groups: biological process (BP), cellular component (CC), and molecular function (MF) genes (Figure 3). Genes containing BP SNPs were involved in various cellular processes, including cellular processes (13,144 genes), metabolic processes (11,154 genes), organic substance metabolic processes (10,584 genes), and cellular metabolic processes (10,038 genes). CC SNPs were detected in genes associated with intracellular anatomical structures (15,349 genes), organelle entities (13,635 genes), and intracellular organelles (13,624 genes). Regarding MF SNPs, the genes were annotated with GO terms related to binding activities, such as binding (9513 genes), organic cyclic compound binding (5795 genes), heterocyclic compound binding (5776 genes), ion binding (4955 genes), and protein binding (3451 genes). This analysis provides insights into the functional consequences of gamma ray irradiation-induced mutations, thus highlighting the diverse biological processes, cellular components, and molecular functions affected by these mutations. 

### 2.5. GWAS Reveals SNPs Associated with Agronomic Traits

Utilizing a generalized linear model, we conducted association analysis to investigate the genetic underpinnings of four key agronomic traits. The analysis of Manhattan and QQ plots (Figure 4) unveiled 76 SNPs distributed across 14 chromosomes significantly associated with thousand-seed weight at a significance threshold level of −log_10_(*p*) = 4.864 (Table 5). However, no suggestive or significant SNPs were identified for plant height, ear length, or seed yield. Out of the 73,226 union SNPs dataset associations examined, 76 SNPs exhibited significant associations with agronomic characteristics when applying the generalized linear model (GLM). Among these selected SNPs, 55 were situated within genic regions, while 21 were detected in intergenic regions. Notably, we annotated a total of 14 genes associated with the thousand-seed weight variable, including S-formylglutathione hydrolase-like (SFGH), zinc finger BED domain-containing protein RICESLEEPER 1-like (ZBED1), midasin-like (MDN1), villin-3-like isoform X2 (VLN3), a death-associated inhibitor of apoptosis 1 (DI-AP1), ubiquitin-like-specific protease 1C (ULP1C), chitinase 1 (CHIT1), amino acid transporter AVT1A (AVT1A), endoribonuclease Dicer homolog 1-like (DLC1), AP-1 complex subunit mu-2 (AP1M2), RNA-binding protein 5 (RBP5), ATPase 6, plasma membrane-type (AHA6), serine acetyltransferase 5 (SAT5), and U2 snRNP-associated SURP motif-containing protein isoform X2 (U2SURP).

### 2.6. GWAS Exposed SNPs Associated with Fatty Acid Compositions and Crude Fat Content

Using a generalized linear model, association analysis was conducted to explore the genetic basis of eight fatty acid compositions and crude fat content levels. Manhattan and QQ plots (Figure 5) highlighted significant associations, with 2 SNPs detected across 1 chromosome for C16:1, 1 SNP across 1 chromosome for C18:0, 1 SNP across 1 chromosome for C18:1, 6 SNPs across 6 chromosomes for C20:1, 26 SNPs across 10 chromosomes for C22:1, and 37 SNPs across 10 chromosomes for crude fat, thus surpassing the significance threshold level (−log_10_(*p*) = 4.864) (Table 6). However, no suggestive or significant SNPs were observed for C16:0, C18:2, or C18:3. Out of the 73,226 union SNPs dataset associations examined, 73 SNPs exhibited significant associations with oil traits when applying the generalized linear model (GLM). Among these selected SNPs, 37 were situated within genic regions, while 36 were detected in intergenic regions. Regarding gene annotations, one gene (encoding a putative E3 ubiquitin ligase (SUD1)) was annotated from C16:1. For C20:1, two SNPs located in the genic region (CDS) were associated with probable Xaa-Pro aminopeptidase P isoform X2 (XPNPEP1) and a hypothetical protein. Additionally, 26 SNPs were significantly associated with C22:1, including six genes: gene family (EXO70), para-myosin-like (Prm), mediator of RNA polymerase II transcription subunit 34 (MED34), cyclin-C1-1 (CCNC), ribonuclease T2 family (RNASET2), and protein ENHANCED DISEASE RESISTANCE 2-like isoform X2 (EDR2)). For the crude fat content, a total of 10 genes (part of a nanodomain complex that tethers PI4Kα1 to the plasma membrane (PI4Kα1), 1-aminocyclopropane-1-carboxylate synthase 6 (ACS6), mitofer-rin (Mfrn), Transmembrane amino acid transporter family protein (TAAT), chromo domain-containing protein LHP1-like (LHP1), nonlysosomal glucosylceramidase (GBA2), RING-H2 finger protein ATL47-like (ATL47), embryogenesis-associated protein EMB8 (EMB8), CEP DOWNSTREAM 1 (CEPD1), and protein CHUP1, chloroplastic-like (CHUP1)) were annotated from crude fat.

## 3. Discussion

Rapeseed breeding has prioritized high seed yield, which is a trait influenced by environmental conditions, genotypes, and their interactions [7]. Previous studies have emphasized the significant impact of genotypes on rapeseed seed yield [7,8]. However, the limited genetic resources available for rapeseed breeding have hindered further improvements in yield potential [8,9,24]. Various agronomic traits, including ear length, branch numbers, silique numbers, number of seeds per silique, thousand-seed weight, and disease resistance, contribute to rapeseed seed yield [2,7,8,12,24]. The original cultivar ‘Tamra’ exhibited a low seed yield (300–350 kg/10a), thus resulting in diminished industrial value [25]. In this study, the original cultivar ‘Tamra’ and 95 rapeseed mutant lines were evaluated over two years for seed yield. Fifteen mutant lines displayed approximately 26% higher yields (over 390 kg/10a) compared to the original cultivar, thus indicating their potential as candidates for developing new rapeseed cultivars with improved seed yield traits.

Industrial demands for rapeseed oils necessitate modifications in fatty acid compositions, which are recognized by breeders [4,13]. These compositions, including oleic acid, linoleic acid, and linolenic acid, influence the value and applications of rapeseed oils [4,6]. Gamma ray irradiation-induced mutations have been effective in altering the fatty acid compositions of various oilseed crops, including rapeseed [14,15,23,26]. In this study, significant changes were observed in oleic acid and erucic acid contents among the mutant lines compared to the original cultivar ‘Tamra’. Mutant lines such as Tr14-1 and Tr14-5 exhibited higher erucic acid contents (≥20%) than other genotypes, thus indicating the influence of gamma ray irradiation-induced mutations on fatty acid compositions in rapeseed. These mutant lines hold promise as materials for developing new rapeseed cultivars with improved oil traits.

The GO analysis of rapeseed genes with polymorphic SNPs induced by gamma irradiation revealed functional changes in mutated genes. In the BP category, mutations were most frequently observed in genes involved in cellular processes (GO:0009987), metabolic processes (GO:0008152), organic substance metabolic processes (GO:0071704), cellular metabolic processes (GO:0044237), and primary metabolic processes (GO:0044238). Regarding the CC category, mutations were associated with intracellular anatomical structures (GO:0005622), organelles (GO:0043226), intracellular organelles (GO:0043229), membrane-bound organelles (GO:0043227), and intracellular membrane-bound organelles (GO:0043231). In terms of the MF category, major polymorphic SNPs were linked to GO terms such as binding (GO:0005488), organic cyclic compound binding (GO:0097159), heterocyclic compound binding (GO:1901363), ion binding (GO:0043167), and protein binding (GO:0005515). Guan et al. (2023) previously reported that the GO enrichment analysis of highly expressed genes during seed development in *B. napus* highlighted the importance of gene expression, translational initiation, and cellular nitrogen compound metabolic processes in the BP category; intracellular anatomical structures, organelles, and intracellular organelles in the CC category; and translation regulator activity, translation factor activity, RNA binding, and nucleic acid binding in the MF category [27]. These findings align with our results, thus confirming the significance of these terms in seed development in rapeseed. Additionally, a previous GO analysis of variation in fatty acid-mutated rapeseed genotypes induced by gamma rays identified similarities in cellular processes, primary metabolic processes, nitrogen compound metabolic processes, intracellular entities, organelles, intracellular organelles, nucleotide binding, nucleoside phosphate binding, and anion binding [6]. This suggests a consistent pattern of mutations induced by radiation mutagenesis across rapeseed mutant lines.

The GWAS identified 14 SNPs significantly associated with the thousand-seed weight variable (−log_10_(*p*) > 4.864). Seed weight, a complex trait influenced by multiple genes and genotypes, plays a pivotal role in seed yield, germination rate, and seedling vigor [7,8,9,11]. Understanding the molecular mechanisms governing seed weight regulation is crucial for plant breeders. Our study revealed associations between the thousand-seed weight variable and the following genes: SFGH, ZBED1, MDN1, VLN3, DCL1, AP1M2, RBP5, AHA6, SAT5, AVT1A, U2SURP, DIAP1, ULP1C, and CHIT1. While the exact role of some genes in seed weight regulation remains unclear, others have been implicated in various cellular processes. For instance, SFGH, encoding a protein belonging to the esterase/lipase/thioesterase family, detoxifies formaldehyde, thus potentially impacting leaf size, biomass, and grain yield in wheat [28]. ZBED1, a transcription factor, regulates seed development and maturation, thus influencing lipid metabolism, hormone signaling, and stress responses [29,30]. MDN1, categorized as a “low-quality protein” gene, regulates seed weight by affecting storage protein accumulation [31]. VLN3, an actin-binding protein, may play a role in seed development and germination [32]. DCL1, involved in RNA interference, affects seed weight by modulating microRNA processing [33]. AP1M2, a transcription factor, regulates seed weight through the JUN-like transcription factor (JUB1), thus promoting seed size via cell proliferation and expansion [34,35]. RBP5, an RNA-binding protein, impacts seed weight by regulating embryo development and carbon metabolism [36,37]. AHA6, a P-type ATPase, influences seed weight by enhancing nutrient uptake and transport to developing seeds [38,39]. SAT5 positively correlates with seed weight, which is likely through increased seed size [40]. AVT1A, an amino acid transporter, affects seed weight by modulating amino acid content [41,42]. U2SURP indirectly regulates seed weight by influencing plant growth and development. Regarding fatty acid composition, the GWAS identified eight SNPs significantly associated with palmitoleic acid, eicosenoic acid, and erucic acid [40]. SUD1, an E3 ubiquitin ligase, potentially regulates palmitoleic acid metabolism [43,44]. XPNPEP1 is linked to eicosenoic acid, although the direct relationship remains unclear. For erucic acid, EXO70, MED34, Prm, CCNC, RNASET2, and EDR2 genes were implicated [45,46,47,48,49]. EXO70, involved in vesicular trafficking, may influence lipid transport and secretion metabolism. EDR2, regulating plant immune responses, likely affects fatty acid biosynthesis via negative regulation. For crude fat content, 10 genes were annotated, including PI4Kα1, ACS6, Mfrn, TAAT, LHP1, GBA2, ATL47, EMB8, CEP D1, and CHUP1 [50,51,52,53,54,55]. These genes are involved in various cellular processes, thus potentially impacting fatty acid biosynthesis. Overall, these findings shed light on the genetic basis of seed weight and fatty acid composition in rapeseed, thus providing valuable insights for crop improvement efforts.

## 4. Materials and Methods

### 4.1. Plant Material

The seed of the ‘Tamra’ cultivar was obtained from the Bioenergy Crop Research Center (Rural Development Administration, Jeonju-si, Republic of Korea). Mutant rapeseed lines were generated by treating the seeds with 700 Gy of gamma (^60^Co) radiation at 2008 [6]. The procedure used to develop mutant rapeseed lines is shown in Figure 6. The treated seeds were sown to obtain the M_1_ generation, and seeds from one silique (developed from the main stem of each M_1_ plant) were harvested. M_2_ seeds from 200 individual plants were grown with a single replicate. In the M_2_ generation, all individuals were investigated for morphological and agronomic mutations relative to the original cultivar. One hundred and twenty rapeseed mutants, selected based on their agronomic characteristics, were obtained from the M_3_ and M_5_ generations. We analyzed the uniformity of the fatty acid compositions by GC-MS (gas chromatography mass spectrometry) and crude fat content for two generations (M_6_ to M_7_) to select stable lines. Finally, 95 rapeseeds mutant lines that varied in agronomic characteristics, fatty acid compositions, and crude fat content and exhibited stable inheritance of the mutated characteristics from M_8_ generations were selected. The selfing procedure was continued until the M_8_ generation. Four agronomic traits, including plant height, ear length, thousand-seed weight, and seed yield, were investigated according to the International Union for the Protection of New Varieties of Plants (UPOV) test guidelines for rapeseed and the standard of research and investigation for agronomic traits. Each agronomic trait was measured with three biological replicates. Also, crude fat-containing fatty acids had three biological replicates, which were randomly sampled and mixed into a single sample. Cultivars with radiation-generated mutant genotypes were grown by the Radiation Breeding Research Team at the Advanced Radiation Technology Institute of the Korea Atomic Energy Research Institute, Korea.

### 4.2. Determination of Fatty Acid Compositions and Crude Fat Contents

The seed oil content was analyzed using the AOAC method. Using the Soxhlet extraction procedure, 5 g crushed seeds (80 mashed) was packed into a thimble, and the oils were extracted with diethyl ether for 6 h. Fatty acid compositions were measured according to the method as previously described. The rapeseed oil was extracted from rapeseed powder in 1 mL of chloroform–hexane–methanol (8:5:2, *v*/*v*/*v*) for 12 h. From this, 200 μL of extracted oil was added to 75 μL of methylation reagent (0.25 M methanolic sodium methoxide: petroleum ether: ethyl ether, 1:5:2, *v*/*v*/*v*) for derivatization. Hexane was added to bring the total volume up to 1 mL. The fatty acid composition of the rapeseed seed oil was analyzed using a GC-MS (Plus-2010, Shimadzu, Kyoto, Japan) instrument equipped with an HP-88 capillary column (J&W Scientific, Folsom, CA, USA, 60 m × 0.25 mm × 0.25 μm) under the following conditions: ionization voltage—70 eV; mass scan range—50–450 mass units; injector temperature—230 °C; detector temperature—230 °C; injection volume—1 μL; split ratio—1:30; carrier gas—helium; and flow rate—1.7 mL/min. The column temperature program specified an isothermal temperature of 40 °C for 5 min in-creasing to 180 °C at a rate of 5 °C/min, followed by a subsequent increase to 230 °C at a rate of 1 °C/min. The substances present in the extracts were identified according to their retention time (RT) and using the mass spectra database (NIST 62 Library).

### 4.3. DNA Extraction

Young leaves were sampled from the original cultivar ‘Tamra’ and 95 rapeseed mutant lines. Genomic DNA was isolated using a DNeasy 250 Plant Mini Kit (Qiagen, Hilden, Germany) according to the manufacturer’s instructions. The extracted DNAs were stored at 4 °C until use. Polymerase chain reaction (PCR) analysis and DNA concentrations were determined using a NanoDrop ND-1000 spectrophotometer (Thermo Fisher Scientific., Waltham, MA, USA) and were then adjusted to 30 ng/μL.

### 4.4. Library Construction and Genotyping-by-Sequencing (GBS) Analysis

GBS libraries were prepared using the restriction enzyme ApeKI (5′-GCWGC-3′; New England Biolabs, MA, USA) following a protocol adapted from a previous study [56]. Oligonucleotides representing the top and bottom strands of each barcode adapter, along with a common adapter, were diluted separately with TE buffer (50 μM each) and annealed using a thermocycler. DNA samples (100 ng/μL) were added to individual wells containing the appropriate adapter. Afterward, the samples (DNA with adapters) underwent overnight digestion with ApeKI at 75 °C. Subsequently, sets of digested DNA samples, each with a distinct barcode adapter, were combined (5 μL each) and purified using the QIAquick PCR Purification Kit (Qiagen, San Diego, CA, USA) as per the manufacturer’s instructions. The resulting restriction fragments from each library were then amplified in 50 μL volumes. These volumes contained 2 μL of pooled DNA fragments, Herculase II Fusion DNA Polymerase (Agilent, Santa Clara, CA, USA), and 25 pmol of each primer: Primer A (5′-AAT GAT ACG GCG ACC ACC GAG ATC TAC ACT CTT TCC CTA CAC GAC GCT CTT CCG ATC T-3′) and Primer B (5′-CAA GCA GAA GAC GGC ATA CGA GAT CGG TCT CGG CAT TCC TGC TGA ACC GCT CTT CCG ATC T-3′). These amplified sample pools constituted the sequencing ‘library’, which was subsequently sequenced on the Illumina HiSeq X Ten platform by SEEDERS Co. (Daejeon, Korea).

### 4.5. Sequence Preprocessing and Alignment to Reference Genome Sequence

Demultiplexing was conducted utilizing the barcode sequence, which was followed by adapter sequence removal and sequence quality trimming. Adapter trimming was carried out using Cutadapt (version 1.8.3) [57], while sequence quality trimming was performed using the DynamicTrim and LengthSort modules of the SolexaQA program (version 1.13) [58]. DynamicTrim was employed to trim low-quality bases at both ends of short reads based on the Phred score, thus refining them into high-quality cleaned reads. Subsequently, LengthSort was applied to remove any excess base cuts introduced by DynamicTrim, with the criterion of a Phred score of DynamicTrim ≥ 20 and LengthSort retaining short-read lengths of ≥ 25 bp. Cleaned reads passing the preprocessing steps were subjected to mapping to the reference genome sequence using BWA (version 0.7.17-r1188) [59]. This mapping served as a preliminary step to identify raw SNPs (single nucleotide polymorphisms) and In/Del (insertion/deletion) sequences within the samples. The resultant BAM format file was generated with default parameter values, except for specific options: a seed length (-l) of 30, maximum differences in the seed (-k) of 1, number of threads (-t) set to 16, mismatch penalty (-M) of 6, gap opening penalty (-O) of 15, and gap extension penalty (-E) of 8. The experiment was conducted in repetition for validation and consistency.

### 4.6. Raw SNP Detection and Consensus Sequence Extraction

Clean reads obtained from the sequencing process were aligned to the standard genome sequence, and the resulting BAM format files were utilized for the identification of raw SNPs using SAMtools (version 0.1.16) [60]. Consensus sequences were then extracted from these alignments. Prior to SNP detection, SNP validation was performed using an in-house script developed by SEEDERS [61]. During raw SNP detection, default parameters were employed, except for specific options: a minimum mapping quality threshold for SNPs (-Q) of 30, a minimum mapping quality threshold for gaps (-q) of 15, a mini-mum read depth threshold (-d) of 3, a minimum indel score for nearby SNP filtering (-G) of 30, SNPs within INT base pairs around a gap to be filtered (-w) of 15, a window size for filtering dense SNPs (-W) of 15, and a maximum read depth threshold (-D) of 675. These parameters were chosen to ensure accurate SNP detection while minimizing false positives.

### 4.7. Generate SNP Matrix

For the analysis of SNPs among the studied subjects, an integrated SNP matrix was generated across samples. Initially, a list of shared SNP positions was created by com-paring each sample with a standard reference genome, with non-SNP loci filled in from the consensus sequence of each sample. Subsequently, the final SNP matrix was constructed by filtering potentially miscalled SNP positions through comparison across samples. SNPs were categorized into homozygous (SNP read depth ≥ 90%), heterozygous (40% ≤ SNP read depth ≤ 60%), and other (homozygous/heterozygous; not distinguishable by type) groups based on their read depth. These SNP positions were then classified as either “intergenic” or “genic regions” based on their location within the standard reference genome sequence. Genic regions were further subdivided into “CDS (coding sequence)” or “intron regions”. In the integrated SNP matrix, priority was given to selecting common SNPs found in the original cultivar ‘Tamra’ when comparing mutant lines. Polymorphic SNPs were identified by comparing the common SNP of the original cultivar with the base sequence of each mutant. To facilitate gene ontology (GO) analysis and explore relationships, SNP loci from each mutant line were integrated to ensure comprehensive coverage of SNP loci across all samples.

### 4.8. Gene Ontology (GO) Analysis of Genes with Polymorphic SNPs

Gene ontology alignment was conducted utilizing candidate sequences containing polymorphic SNPs, along with sequences obtained from the GO database through in-house scripts [62]. Thresholds were categorized into three functional categories: BP (biological process), CC (cellular component), and MF (molecular function), with a significance level set at 0.01 (E-value ≤ 1.0 × 10^−10^, best hits). This classification ensured the accurate annotation of SNPs based on their putative functional roles within biological processes, cellular components, and molecular functions.

### 4.9. GWAS with Agronomic Characteristics, Fatty Acid, and Crude Fat

For the GWAS, a total of 73,226 filtered SNPs, with a minor allele frequency greater than 5% and missing data less than 30%, were extracted from the raw dataset. These SNPs were utilized for GWAS analysis employing the generalized linear model (GLM) in TASSEL, specifically TASSEL 5 [63]. Default parameter settings were employed for the GWAS analysis. To establish significance thresholds for the −log_10_(*p*) values in the quantile–quantile plot and Manhattan plot, the Bonferroni method was utilized (p = α/n). With 73,226 SNPs considered in this study at a significance level (α) of 1, the Bonferroni-corrected thresholds for the p values were determined as 1.36 × 10^−5^. Consequently, the corresponding −log_10_(*p*) value for the suggestive threshold was calculated as 4.864 [64]. These thresholds provided guidance for identifying SNPs significantly associated with the trait under investigation. Descriptive statistics and correlation analysis were performed using SPSS 27 (IBM, Armonk, NY, USA).

## 5. Conclusions

In this study, a comprehensive analysis of four agronomic traits, eight fatty acid compositions, and crude fat content across the original rapeseed cultivar ‘Tamra’ and 95 mutant lines was investigated. Significant variations were observed among each trait. Leveraging a genomewide association study (GWAS) employing 73,226 filtered SNPs obtained from GBS data, we pinpointed 32 candidate genes significantly associated with the thousand-seed weight, along with three fatty acid compositions (C16:1, C20:1, and C22:1) and crude fat content. Moving forward, to fortify the genetic underpinnings of the thousand-seed weight, fatty acid composition, and crude fat content in rapeseed, it is imperative to conduct functional validation studies on the identified candidate genes. Our findings provide valuable insights into the genetic architecture of these traits, thus offering potential avenues for informing future rapeseed breeding strategies.

## Figures and Tables

**Figure 1 plants-13-01576-f001:**
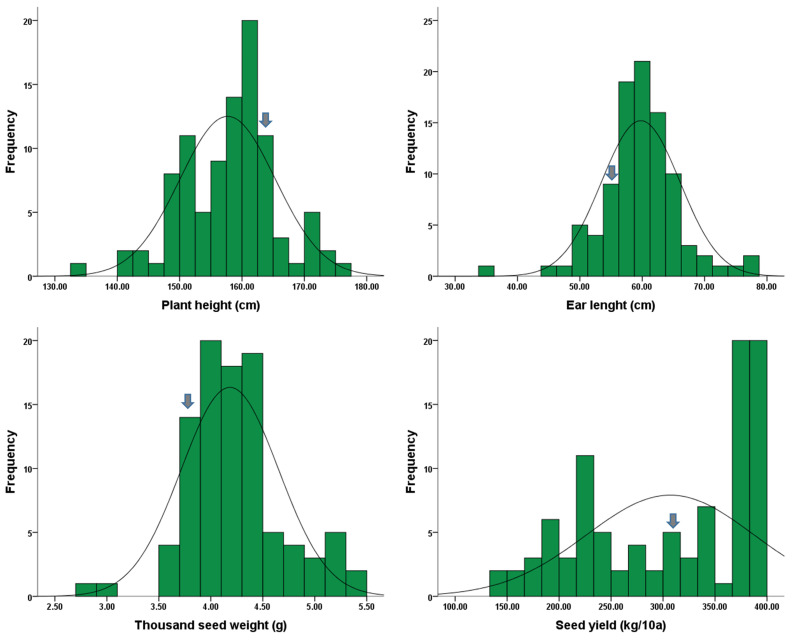
Frequency distribution of agronomic characteristics in 96 rapeseeds. The arrow indicates the original cultivar.

**Figure 2 plants-13-01576-f002:**
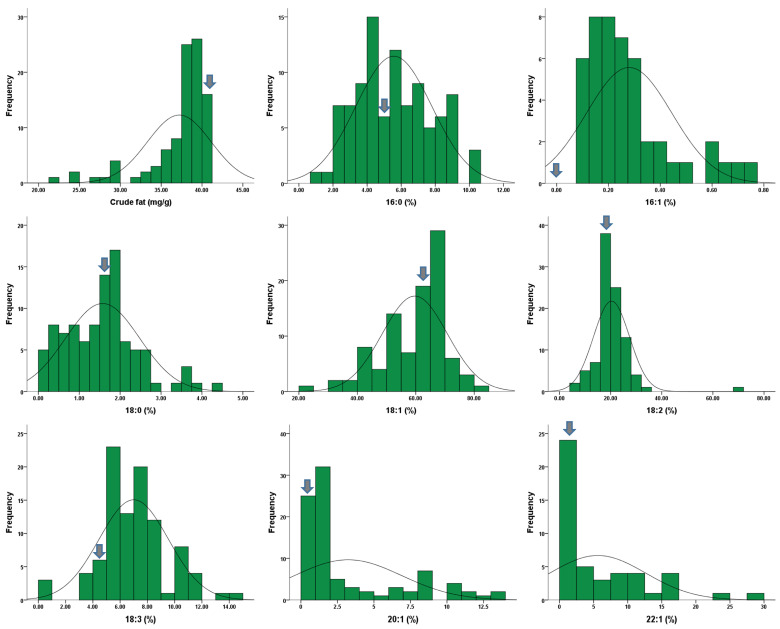
Frequency distribution of crude fat and fatty acid composition in 96 rapeseeds. The arrow indicates the original cultivar.

**Figure 3 plants-13-01576-f003:**
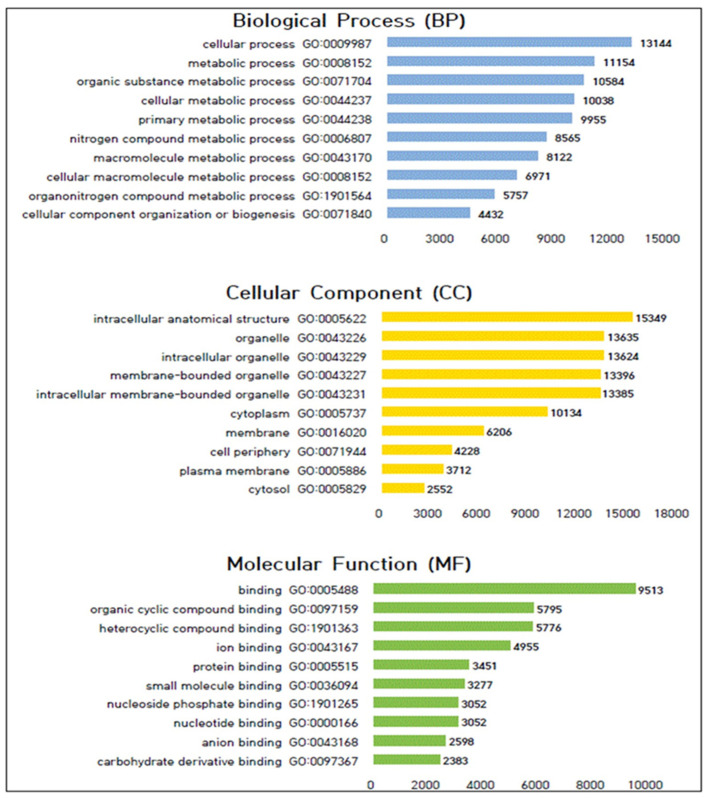
Histogram of GO terms of union SNPs in rapeseed mutant lines.

**Figure 4 plants-13-01576-f004:**
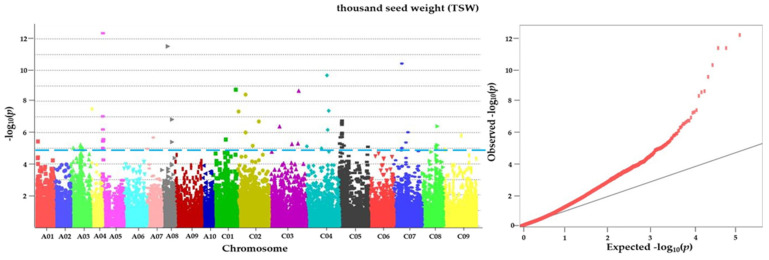
Manhattan plots and quantile–quantile (QQ) plots for thousand-seed weight in original 96 rapeseeds. In the Manhattan plots, the blue line indicates the genomewide threshold −log_10_(*p*) = 4.864, which was calculated using the Bonferroni method.

**Figure 5 plants-13-01576-f005:**
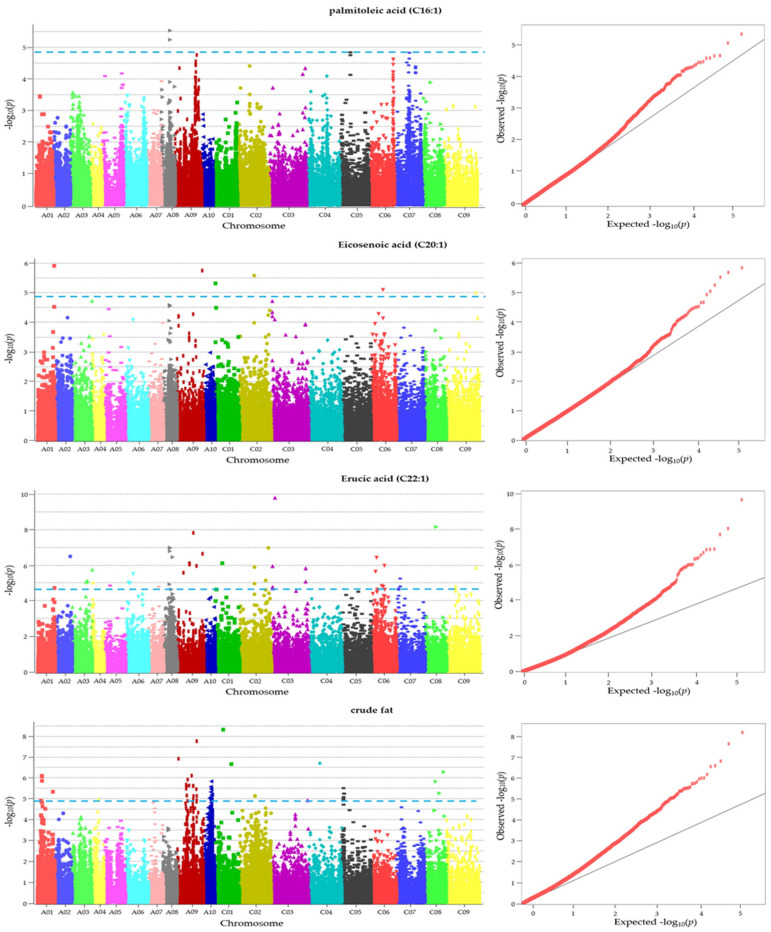
Manhattan plots and quantile–quantile (QQ) plots for 3 fatty acid and crude fat in 96 rapeseeds. In the Manhattan plots, the blue line indicates the genomewide threshold −log_10_(*p*) = 4.864, which was calculated using the Bonferroni method.

**Figure 6 plants-13-01576-f006:**
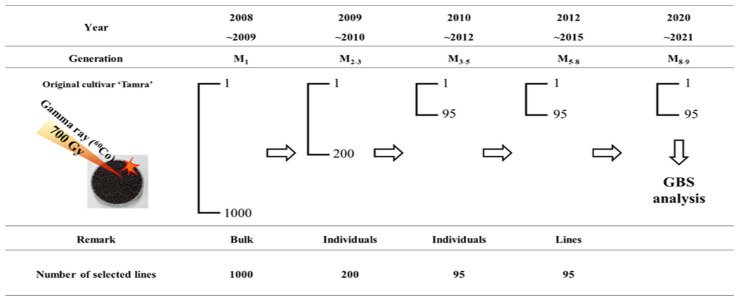
The development of mutant lines. Mutations were derived by irradiating 700 Gy of gamma rays to Korean rapeseed ‘Tamra’, and 95 rapeseed mutant lines with changes in fatty acid compositions, crude fat contents, and seed yield were identified from M_5_ to M_8_ populations. This mutant line was homozygous from the M_5_ to the M_8–9_ generations.

**Table 1 plants-13-01576-t001:** Descriptive statistics for agronomic characteristics in 96 rapeseeds.

Trait	Min	Max	SD	Mean	CV(%)	Skew	Kur
Plant height (cm)	134.00	175.00	7.62	157.69	4.83	−0.23	0.34
Ear length (cm)	35.00	77.50	6.27	59.76	10.49	−0.24	2.78
Thousand-seed weight (g)	2.80	5.40	0.47	4.18	11.15	0.33	1.04
Seed yield (kg/10a)	144.00	398.00	80.26	306.93	26.15	−0.45	−1.28

Min: minimum; Max: maximum; SD: standard deviation; CV: coefficient of variation; Skew: skewness; Kur: kurtosis.

**Table 2 plants-13-01576-t002:** Descriptive statistics for crude fat and fatty acid composition in 96 rapeseeds.

Trait	Min	Max	SD	Mean	CV(%)	Skew	Kur
Crude fat (mg/g)	21.31	40.98	3.88	37.22	10.43	−2.19	4.76
C16:0 (%)	0.88	10.50	2.22	5.57	39.83	0.21	−0.71
C16:1	0.10	0.76	0.16	0.28	58.63	1.41	1.36
C18:0	0.15	4.42	0.90	1.57	57.40	0.68	0.61
C18:1	20.93	81.73	11.07	59.65	18.55	−0.84	0.84
C18:2	7.66	71.09	7.01	20.34	34.46	3.92	27.86
C18:3	0.43	14.68	2.53	7.00	36.17	0.25	0.97
C20:1	0.40	13.10	3.65	3.23	113.14	1.37	0.45
C22:1	0.08	29.12	6.94	5.61	123.83	1.53	1.95

Min: minimum; Max: maximum; SD: standard deviation; CV: coefficient of variation; Skew: skewness; Kur: kurtosis. C16:0—palmitic acid, C16:1—palmitoleic acid, C18:0—strearic acid, C18:1—oleic acid, C18:2—linoleic acid, C18:3—linolenic acid, C20:1—eicosenoic acid, C22:1—erucic acid.

**Table 3 plants-13-01576-t003:** Correlation analysis among the fatty acid composition of 96 rapeseeds.

	C16:0	C16:1	C18:0	C18:1	C18:2	C18:3	C20:1
C16:1	0.588 **						
C18:0	0.625 **	0.268					
C18:1	−0.311 **	−0.54 **	−0.236 *				
C18:2	0.443 **	0.299 *	0.210 *	−0.169			
C18:3	0.420 **	0.377 **	0.332 **	−0.569 **	0.253 *		
C20:1	−0.292 **	0.650	−0.234 *	−0.639 **	−0.235 *	0.007	
C22:1	−0.379 **	0.311	−0.312 *	−0.633 **	−0.255	0.125	0.871 **

*: Significant at *p* ≤ 0.05; **: significant at *p* ≤ 0.01. C16:0—palmitic acid, C16:1—palmitoleic acid, C18:0—strearic acid, C18:1—oleic acid, C18:2—linoleic acid, C18:3—linolenic acid, C20:1—eicosenoic acid, C22:1—erucic acid.

**Table 4 plants-13-01576-t004:** Summary of GBS sequence data and alignment to the reference genome sequence.

	Total	Average/Plant
	Raw data	
Reads	715,259,878	7,450,624
Bases (bp)	108,004,241,578	1,125,044,183
	After trimming	
Reads	655,243,166	6,825,450
Bases (bp)	74,621,891,073	777,311,365
	Mapped reads on reference genome	
Reads	651,288,040	6,784,250
Bases (bp)	3,163,629,539	32,954,474
	Reference genome coverage (%)	3.57%

**Table 5 plants-13-01576-t005:** Annotated genes list of significant associated SNPs with thousand-seed weight in rapeseed.

Chr	Position	−log_10_(*p*)	Gene ID	Description	TAIR ID	Allele
A05	1,312,542	12.37	BnaA05p02350D	*S*-formylglutathione hydrolase-like	AT2G41530.1	A/G
C01	45,614,214	8.75	BnaC01p49650D	zinc finger BED domain-containing protein RICESLEEPER 1-like	AT3G14800.2	G/T
C02	16,544,482	8.44	BnaC02p22490D	midasin-like	AT1G67120.2	G/A
C08	30,501,410	6.41	BnaC08p32830D	villin-3-like isoform X2	AT3G57410.9	A/G
C03	21,138,185	6.40	BnaC03p36140D	death-associated inhibitor of apoptosis 1	AT4G03965.1	C/T
C05	4,473,754	5.90	BnaC05p08830D	ubiquitin-like-specific protease 1C	AT1G10570.3	G/T
A05	2,029,410	5.58	BnaA05p03850D	chitinase 1	AT2G43570.1	T/A
A05	1,197,170	5.46	BnaA05p02100D	amino acid transporter AVT1A	AT2G41190.2	A/C
C05	439,211	5.30	BnaC05p00900D	endoribonuclease Dicer homolog 1-like	AT1G01040.2	T/C
C05	4,629,489	5.25	BnaC05p09110D	AP-1 complex subunit mu-2	AT1G10730.1	C/T
C08	28,339,798	5.12	BnaC08p29480D	RNA-binding protein 5	AT3G54230.5	G/A
A03	20,827,042	5.07	BnaA03p44720D	ATPase 6, plasma membrane-type	AT2G07560.1	T/A
A03	5,465,396	5.02	BnaA03p13200D	serine acetyltransferase 5	AT5G56760.1	G/A
C09	4,097,062	4.91	BnaC09p07010D	U2 snRNP-associated SURP motif-containing protein isoform X2	AT5G25060.1	T/G

**Table 6 plants-13-01576-t006:** Annotated genes list of significant associated SNPs with oil traits in rapeseed.

Trait	Chr_Position	−log_10_(*p*)	Gene ID	Description	TAIR ID	Allele
C16:1	A08_16,580,415	5.24	BnaA08p17150D	Encodes a putative E3 ubiquitin ligase	AT4G34100.1	C/T
C20:1	C01_1,257,401	5.32	BnaC01p02330D	probable Xaa-Pro aminopeptidase P isoform X2	AT4G36760.2	C/A
C22:1	C03_6,878,724	9.78	BnaC03p14350D	EXO70 gene family	AT5G58430.1	C/T
C22:1	C08_21,249,315	8.15	BnaC08p19880D	paramyosin-like	AT1G19980.1	C/T
C22:1	A08_12,298,481	6.98	BnaA08p10580D	mediator of RNA polymerase II transcription subunit 34	AT1G31360.3	C/A
C22:1	C02_56,355,748	6.97	BnaC02p56750D	cyclin-C1-1	AT5G48640.1	A/G
C22:1	C02_51,025,897	5.14	BnaC02p51020D	ribonuclease T2 family	AT2G02990.1	G/A
C22:1	A08_11,469,852	4.92	BnaA08p09630D	protein ENHANCED DISEASE RESISTANCE 2-like isoform X2	AT5G35180.4	G/A
Crude fat	A09_3,387,964	6.92	BnaA09p05630D	Part of a nanodomain complex that tethers PI4Kα1 to the plasma membrane	AT5G26850.1	C/A
Crude fat	A09_18,251,187	5.59	BnaA09p27860D	1-aminocyclopropane-1-carboxylate synthase 6	AT4G11280.1	A/G
Crude fat	C05_2,572,075	5.51	BnaC05p05120D	mitoferrin	AT1G07030.1	G/C
Crude fat	A09_38,265,711	5.46	BnaA09p39540D	Transmembrane amino acid transporter family protein	AT1G25530.1	C/T
Crude fat	A10_15,664,715	5.20	BnaA10p21730D	chromo domain-containing protein LHP1-like	AT5G17690.3	T/C
Crude fat	A09_32,798,569	5.14	BnaA09p32690D	nonlysosomal glucosylceramidase	AT1G33700.1	T/G
Crude fat	A09_39,092,505	5.14	BnaA09p40920D	RING-H2 finger protein ATL47-like	AT1G23980.1	C/T
Crude fat	A09_32,277,805	4.98	BnaA09p32130D	embryogenesis-associated protein EMB8	AT1G34340.1	T/C
Crude fat	C05_2,503,939	4.97	BnaC05p04980D	CEP DOWNSTREAM 1	AT1G06830.1	A/G
Crude fat	C03_71,451,788	4.93	BnaC03p90410D	protein CHUP1, chloroplastic-like	AT1G52080.1	A/T

C16:1—palmitoleic acid, C20:1—eicosenoic acid, C22:1—erucic acid.

## Data Availability

The original contribution presented in the study is publicly available.

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
