# Peer review of "Genotyping-by-Sequencing Analysis Reveals Associations between Agronomic and Oil Traits in Gamma Ray-Derived Mutant Rapeseed (*Brassica napus* L.)"

_plants, 2024, doi:10.3390/plants13111576_

Round 1

Reviewer 1 Report

Comments and Suggestions for Authors

Currently, the analysis of this submission is on the surface. More depth analysis need to be done.

L138. The labels of x-axis and y-axis, especially for numbers, in Figure 1 are too small to be conveniently read.

L235. Authors listed the significant associated SNPs with thousand seed weight in Table. Do these SNPs are non-synonymous substitution?

The analysis is not deep enough. As we know, the Gamma-ray derived mutant contained synonymous substitution and non-synonymous substitution. Besides, some allele could result in stop codon. Authors should associate the phenotype with genotype that the alleles indeed change the genes. Furthermore, although the synonymous substitution wouldn’t change the coded protein, it could alter the expression level or the translation efficiency of mRNA, which should be also take into account. Summarily, to increase the significant, depth, reliability, synonymous substitution and non-synonymous substitution, expression level or the translation efficiency of mRNA, were suggested explored to explain the phenotypes changes.

Author Response

Response to Reviewer 1 Comments

Thank you so much for your suggestion. After considering this suggestion and reviewing the manuscript, we have amended the manuscript to your suggestion.

My comments and questions to authors are as follows:

Point 1: L138. The labels of x-axis and y-axis, especially for numbers, in Figure 1 are too small to be conveniently read.

Response 1: I rewrote the text in that figure with a larger font size.

Point 2: L235. Authors listed the significant associated SNPs with thousand seed weight in Table. Do these SNPs are non-synonymous substitution?.                

Point 3: The analysis is not deep enough. As we know, the Gamma-ray derived mutant contained synonymous substitution and non-synonymous substitution. Besides, some allele could result in stop codon. Authors should associate the phenotype with genotype that the alleles indeed change the genes. Furthermore, although the synonymous substitution wouldn’t change the coded protein, it could alter the expression level or the translation efficiency of mRNA, which should be also take into account. Summarily, to increase the significant, depth, reliability, synonymous substitution and non-synonymous substitution, expression level or the translation efficiency of mRNA, were suggested explored to explain the phenotypes changes.

Response 2 and 3 : As the reviewer mentioned, base substitutions such as non-synonymous and synonymous substitutions cause changes in amino acid sequence that can have a significant impact on trait expression, especially if they have a large effect on protein structure, such as the formation of a stopcodon. GWAS analysis takes a genetic resource and statistically detects the location of SNPs associated with a target trait.

In other words, since GWAS analyses detect common locations where variants that may be associated with a trait exist in multiple genetic resources, the SNPs at those locations may have different substitutions depending on the genetic resource. In this study, we selected candidate genes by GWAS analysis, so the reviewer may think that the depth of the analysis is not enough, but we are planning an experiment to select the lines with the largest difference for traits thousand seed weight, crude fat , fatty acid 16:1, 20:1 and 22:1 detected in the GWAS analysis, check the expression level of the genes by RNA-seq and qRT PCR, and check what kind of substitution has occurred, and will submit it in a next paper.

Again, we appreciate the reviewer's comments.

Reviewer 2 Report

Comments and Suggestions for Authors

In the manuscript “Genotyping-by-sequencing analysis reveals associations between agronomic and oil traits in Gamma-ray-derived mutant Rapeseed (BRASSICA NAPUS L.)”, the authors utilized genotyping-by-sequencing (GBS) to analyze the 73,226 SNPs generated from original cultivar and mutant lines. Notably, 320 SNPs and several genes were implicated in agronomic and oil traits formation. The manuscript is meaningful and can help breeding aimed at improve yield and oil composition in rapeseed. However, there are some concerns that need to be solved before publication.

Major

1. The authors indicated that the mutant lines were generated from gamma ray radiation. Then the authors should take more concentration on the importance of gamma radiation in inducing mutants, since there are novel methods to generate mutant lines after 2008.

2. The authors indicated that 700 Gy of gamma (60Co) radiation was used to trigger mutant lines. Then why they selected this dosage? Had the authors carry out some preliminary experiments?

3. I am aware that the authors have published a work using these mutants (2021, Agronomy). In the work, some SNPs associated with flowering times, crude fat and fatty acid composition in rapeseed were discovered. Then what’s the difference between this work and the published manuscript. Did they have some repetitions? The authors should provide more explanations.

Minor

1. How many replicates were used for determination of crude fat content as well as some other agronomic traits for each line?

2. “BRASSICA NAPUS L.” should be “Brassica napus L.” in the title.

3. Lines 382-383: were measured according to the method as previously described.

4. Lines 490-492: add “was investigated” after the sentence?

Author Response

Response to Reviewer 2 Comments

Thank you so much for your suggestion. After considering this suggestion and reviewing the manuscript, we have amended the manuscript to your suggestion.

My comments and questions to authors are as follows:

Major

Point 1: The authors indicated that the mutant lines were generated from gamma ray radiation. Then the authors should take more concentration on the importance of gamma radiation in inducing mutants, since there are novel methods to generate mutant lines after 2008.

Response 1: As the reviewer comments, Added a sentence to line 87-88 and 90-91 stating that 60% of mutant plants worldwide are caused by physical radiation, of which gamma-ray is known to cause base substitutions, indels, copy number variation, and presence/absence variation due to the breakdown of the double helix structure of DNA.

Point 2: The authors indicated that 700 Gy of gamma (60Co) radiation was used to trigger mutant lines.

Then why they selected this dosage? Had the authors carry out some preliminary experiments?              

Response 2: The authors' organization, the Korea Atomic Energy Research Institute, has a unique gamma-ray facility that can induce plant mutations, so they can easily use different doses of gamma radiation to study mutations. The mutant lines used in this study were derived by irradiation with 700 Gy of gamma radiation. We wanted to increase the mutation rate by using a strong dose, so a large amount of seeds were used and genetic fixation was carried out over several years of selection and analysis.

Point 3: I am aware that the authors have published a work using these mutants (2021, Agronomy). In the work, some SNPs associated with flowering times, crude fat and fatty acid composition in rapeseed were discovered. Then what’s the difference between this work and the published manuscript. Did they have some repetitions? The authors should provide more explanations.

Response 3: (2021, Agronomy) was created by irradiating “Tammi” with 500 Gy of gamm-ray, while the mutant lines used in this study was created by irradiating “Tamra” with 700 Gy of gamm-ray.

Therefore, it is possible to show different spectra for each trait because the raw material used is different, and also for each trait, this paper can be considered a new experiment and result because SNPs that overlap with the results of previous papers that used “Tammi” were not detected, but new SNPs were detected.

Minor

Response 1~4 : Revised the manuscript based on the reviewer's comments, changing the title to “Genotyping-by-sequencing analysis reveals associations be-tween agronomic and oil traits in Gamma-ray-derived mutant Rapeseed (Brassica napus L.)” and adding sentences on lines 387-388 and 496-497 to read “Fatty acid compositions were measured according to the method as previously described” and “In this study, a comprehensive analysis of four agronomic traits, eight fatty acid compositions, and crude fat content across the original rapeseed cultivar ’Tamra' and 95 mutant lines was investigated”.

Also added the sentence “Each agronomic trait was measured with three biological replicates. Also, Crude fat containing fatty acids had three biological replicates, which were randomly sampled and mixed into a single sample” for replicate to lines 374-376.

Again, we appreciate the reviewer's comments.

Reviewer 3 Report

Comments and Suggestions for Authors

1. Why did you choose the GLM model over MLM, given that MLM gives better results?

2. You mention nothing about the population structure of your cultivars in the article, but population structure is involved in the GML model. What software did you use to define it?

3. Why didn't you use the BLUE values ​​for the GWAS analysis to compare with them for a markers match?

4. You have not indicated the software you used for statistical data processing.

Author Response

Point 1: Why did you choose the GLM model over MLM, given that MLM gives better results?

Response 1: In general, GWAS is used to analyze various genetic resources. As the reviewer mentioned, MLM can produce better results than GLM because it uses population structure and kinship to control the false positive rate. However, in this study, we chose GLM, which does not apply kinship, because we analyzed a lines created by treating one variety with a mutagen. We performed both MLM and GLM as shown in Figure 1, and we used GLM because the results were better in GLM than MLM.

Figure 1. GWAS result of MLM and GLM

Point 2: You mention nothing about the population structure of your cultivars in the article, but population structure is involved in the GML model. What software did you use to define it?

Response 2: Phylogenetic tree with 1000 replicates bootstrap using filtered SNPs was constructed with Mega 6. Two independent and sub-related groups were found, but no characteristic subdivision of the related group was found, as shown in figure 2, so it was not included in the data.

Figure 2. Phylogenetic tree

Point 3: Why didn't you use the BLUE values ​​for the GWAS analysis to compare with them for a markers match?.

Response 3: In GWAS analysis, calculating the R2 value using BLUP can measure the additional genetic variance explained by the marker, while BLUE is the amount of phenotypic variance explained by the marker.

Although BLUP has the disadvantage of being compressed toward the mean, we did not use BLUE in this study because the mutant lines were selected from the parental lines and have been fixed and small in variation as they progressed through the M9 generation, and we did not want confounding signals from environmental genotype effects.

Point 4: You have not indicated the software you used for statistical data processing.

Response 3: Statistical analyses were performed using SPSS 27 and are added at line 493-494 of the manuscript.

Round 2

Reviewer 1 Report

Comments and Suggestions for Authors

Nice work for the revision.

Reviewer 2 Report

Comments and Suggestions for Authors

After careful reading, my concerns about some points in the manuscript was successfully solved.